# Cryptic Diversity and Demographic Expansion of *Plasmodium knowlesi* Malaria Vectors in Malaysia

**DOI:** 10.3390/genes14071369

**Published:** 2023-06-28

**Authors:** Sandthya Pramasivan, Van Lun Low, Nantha Kumar Jeyaprakasam, Jonathan Wee Kent Liew, Romano Ngui, Indra Vythilingam

**Affiliations:** 1Department of Parasitology, Faculty of Medicine, Universiti Malaya, Kuala Lumpur 50603, Malaysia; sandthya96@gmail.com (S.P.); nromano@unimas.my (R.N.); 2Tropical Infectious Diseases Research & Education Centre (TIDREC), Universiti Malaya, Kuala Lumpur 50603, Malaysia; 3Biomedical Science Program, Center for Toxicology and Health Risk Studies, Faculty of Health Sciences, Universiti Kebangsaan Malaysia, Kuala Lumpur 50300, Malaysia; nanthakumar@ukm.edu.my; 4Environmental Health Institute, National Environment Agency, Singapore 569874, Singapore; jonathan_liew@nea.gov.sg; 5Malaria Research Centre, Faculty of Medicine and Health Sciences, Universiti Malaysia Sarawak (UNIMAS), Kota Samarahan 94300, Sarawak, Malaysia

**Keywords:** *Plasmodium knowlesi*, Malaysia, *Anopheles*, mosquitoes, genetic diversity

## Abstract

Although Malaysia is considered free of human malaria, there has been a growing number of *Plasmodium knowlesi* cases. This alarming trend highlighted the need for our understanding of this parasite and its associated vectors, especially considering the role of genetic diversity in the adaptation and evolution among vectors in endemic areas, which is currently a significant knowledge gap in their fundamental biology. Thus, this study aimed to investigate the genetic diversity of *Anopheles balabacensis*, *Anopheles cracens*, *Anopheles introlatus*, and *Anopheles latens*—the vectors for *P. knowlesi* malaria in Malaysia. Based on cytochrome c oxidase 1 (*CO1*) and internal transcribed spacer 2 (*ITS2*) markers, the genealogic networks of *An. latens* showed a separation of the haplotypes between Peninsular Malaysia and Malaysia Borneo, forming two distinct clusters. Additionally, the genetic distances between these clusters were high (2.3–5.2% for *CO1*) and (2.3–4.7% for *ITS2*), indicating the likely presence of two distinct species or cryptic species within *An. latens*. In contrast, no distinct clusters were observed in *An. cracens*, *An. balabacensis*, or *An. introlatus*, implying a lack of pronounced genetic differentiation among their populations. It is worth noting that there were varying levels of polymorphism observed across the different subpopulations, highlighting some levels of genetic variation within these mosquito species. Nevertheless, further analyses revealed that all four species have undergone demographic expansion, suggesting population growth and potential range expansion for these vectors in this region.

## 1. Introduction

Malaria remains a persistent global public health challenge, and countries in Southeast Asia have been assigned the goal of malaria elimination by 2030. This ambitious target highlights the urgency and importance of concerted efforts to combat malaria and reduce its burden in the region. Malaysia has been free of human malaria since 2018 [1], but *P. knowlesi*, a simian malaria parasite, is the predominant species currently occurring in the country [2]. All countries in SEA have reported the occurrence of *P. knowlesi*, with the exception of Timor-Leste [3]. It is crucial to consider the WHO [4] recommendation to postpone the certification of a malaria-free status for countries reporting significant *P. knowlesi* cases in the region. This highlights the importance of ongoing surveillance, monitoring, and control efforts to effectively address the persistence of malaria and prevent the potential reintroduction of human malaria in Malaysia and neighboring countries.

In addition to *P. knowlesi*, other simian malarias, such as *Plasmodium cynomolgi*, *Plasmodium inui*, and *Plasmodium fieldi*, have been reported in Southeast Asia [5,6,7,8,9,10,11,12,13,14]. The long-tailed (*Macaca fascicularis*) and pig-tailed macaques (*Macaca nemestrina*) are the primary hosts of these simian malaria parasites. Recent studies have shown that *P. cynomolgi* and *P. inui* are the predominant species occurring in macaques [15]. Similar findings have been observed in the simian malaria vectors [3].

*Anopheles hackeri* (belonging to the Leucosphyrus Group) was the first species to be incriminated as the vector of *P. knowlesi* in Peninsular Malaysia [16]. This was followed by the incrimination of *An. latens* [17,18], *An. cracens* [19,20], *An. balabacensis* [21], and *An. introlatus* [22], which all belong to the Leucosphyrus Group of mosquitoes, as vectors for *P. knowlesi* in Malaysia. With changes in landscape and deforestation, humans, macaques, and mosquitoes are now found in the same environment, thus enabling the transmission of simian malaria to humans [23].

To obtain certification as a malaria-free country, it is crucial to address the spread of simian malaria within Malaysia. However, there are currently limited data on the vectors responsible for transmitting simian malaria in the country. The Leucosphyrus Group of *Anopheles*, which comprises a complex of species, presents challenges in morphological identification of each species. Additionally, there is a lack of comprehensive information on the genetic studies of natural vectors of simian malaria in Malaysia. In a previous study, the genetic variation within the subpopulations of *An. balabacensis* in Kudat, Sabah, was examined using mitochondrial genes, revealing an expanding and growing population of *An. balabacensis* in Sabah. Notably, while the overall population showed low genetic diversity, the subpopulations exhibited high genetic diversity, likely due to interpopulation migration and breeding, facilitating gene flow among the subpopulations [24].

Hence, the information on genetic diversity of vectors in endemic areas is crucial for determining species taxonomy and the spatial limitations of natural populations [25]. Based on this knowledge, researchers may understand and predict the epidemiology, distribution, and transmission dynamics of vector-borne diseases [25]. As a result, studies on the genetic diversity and population structure of malaria vectors are critical for successfully executing vector control programs against malaria in the country.

Thus, this study aimed to analyze the population genetic structure and genetic diversity of the four important malaria vectors, *An. balabacensis*, *An. cracens*, *An. introlatus*, and *An. latens*, using the mitochondrial and ribosomal sequences for the first time in Malaysia.

## 2. Materials and Methods

### 2.1. Study Location

*Anopheles* mosquitoes were collected from multiple states across Malaysia, including Negeri Sembilan, Johor, Kelantan, Pahang, Perak, and Sarawak. This sampling strategy ensured that the study covered the north, south, east, and west regions of the country. The selection of survey locations was based on the Ministry of Health (MoH), Malaysia’s data on *P. knowlesi* malaria cases, allowing for a comprehensive assessment of vector populations in areas with known incidences of the disease (Figure 1).

### 2.2. Sample Collection

Mosquitoes were collected using bare-leg catch (BLC), human-baited trap, Mosquito Magnet, and CDC light trap [26] during the sampling period from June 2019 until January 2021 between 1800 and 0000 h. The sample collection process was mainly through the BLC method, whereas other sampling methods depended on the specific location and the workforce available for the study. Detailed information on the samples from the different sampling locations is illustrated in Table 1.

### 2.3. DNA Extraction and Polymerase Chain Reaction

DNA was extracted from the mosquitoes’ legs using InstaGene Matrix (Bio-Rad, Hercules, CA, USA) according to the manufacturers’ protocol. The extracted DNA was kept at −20 °C until required. All *Anopheles* mosquitoes from the Leucosphyrus Group obtained in this study, including some archived samples, were further molecularly characterised using the internal transcribed spacer 2 (*ITS2*) region and mitochondrial cytochrome c oxidase subunit 1(*CO1*) gene. The *ITS2* was amplified by ITS2A and ITS2B primers [27], with the PCR conditions as follows: denaturation at 95 °C for 2 min; 35 cycles of amplification at 95 °C for 30 s; annealing step at 51 °C for 30 s, with elongation step at 72 °C for 1 min; followed by final elongation step of 10 min at 72 °C. LCO1490 and HCO2198 primers [28] were used to amplify the *CO1* gene. The PCR conditions were as follows: denaturation at 95 °C for 3 min; 35 cycles of amplification at 95 °C for 1 min; annealing step at 50 °C for 1 min, with elongation step at 72 °C for 1 min; followed by final elongation step of 10 min at 72 °C and held at a temperature of 4 °C. Each reaction mixture of 25 μL contained 5 μL DNA template, 0.5 μM primers, respectively, 0.2 mM dNTP, 3 mM MgCl2, 1 × GoTaq^®^ Flexi Buffer, and 1.0 U of GoTaq^®^ DNA polymerase (Promega Corporation, Madison, WI, USA). This master mix was used for both primer sets. Amplicons were subjected to electrophoresis on 1.5% agarose gels. The amplified product was purified from the gel and outsourced for Sanger sequencing (Apical Scientific Sdn. Bhd., Malaysia). Sequences of each species were performed using the Basic Local Alignment Search Tool (BLAST) (http://blast.ncbi.nlm.nih.gov/Blast.cgi, accessed on 12 February 2021) for similarity searches. A species was confirmed by ≥98% identity and query coverage to the deposited sequence.

### 2.4. Data Analyses for Genetic Studies

The study included sequences of mosquitoes collected from Peninsular Malaysia and Sarawak, while additional sequences from Sabah and Selangor were retrieved from the NCBI GenBank, providing a comprehensive representation of mosquito populations across different regions of Malaysia. *CO1, ITS2*, and combined sequences of each species were aligned using BioEdit (Version 7.2) [29]. Haplotype networks for *An. balabacensis*, *An. cracens*, *An. introlatus*, and *An. latens*, based on their polymorphic sites, were constructed by using the median-joining method in NETWORK version 5.0.0.1 software (Fluxus Technology LTD, Suffolk, UK). The number of haplotypes in the subpopulations, haplotype diversity [30], and nucleotide diversity [31] were also estimated using DnaSP 5.0 software. Genetic distances among species/populations were calculated using MEGA 11 software [32].

Pairwise genetic differentiation (*F_ST_*) and gene flow (Nm) values between the subpopulations of each species were tested for significance. DnaSP 5.0 software was also used in estimating gene flow using 1000 permutations [33]. The levels of genetic differentiation can be categorized as *F_ST_* > 0.25 (great differentiation), 0.05 to 0.25 (moderate differentiation), and *F_ST_* < 0.05 (negligible differentiation) [34]. The levels of gene flow can be categorized as Nm > 1 (high gene flow), 0.25 to 0.99 (intermediate gene flow), and Nm < 0.25 (low gene flow) [35]. To analyse the randomness of DNA sequence evolution, a neutrality test was performed by using Tajima’s D [36] and Fu’s Fs [37] with 1000 simulations. Mismatch analysis, Harpending’s raggedness index (Rag) [38], and the R_2_ statistic of Ramos-Onsins and Rozas [39] were used to investigate demographic expansion.

## 3. Results

### 3.1. Genetic Diversity Indices

*An. introlatus* had a greater number of haplotypes in the combined sequences (n = 25), followed by *An. latens* (n = 15) and *An. cracens* (n = 7), and the least was *An. balabacensis* (n = 5) (Table 1). The *An. introlatus* subpopulations from Kongsi Balak, Johor (Hd = 0.899 ± 0.031, π = 0.00273 ± 0.00045), *An. latens* from Rumah Sewa Panto, Sarawak (Hd = 1.000 ± 0.177, π = 0.01410 ± 0.00642), *An. cracens* from Kem Sri Gading, Pahang (Hd = 0.791 ± 0.044, *p* = 0.00093 ± 0.00009), and *An. balabacensis* from Simpang Utong, Sabah (Hd = 1.000 ± 0.27, *p* = 0.00168 ± 0.00059) had the highest haplotype and nucleotide diversities.

*An. cracens* had the least number of haplotypes in *ITS2* (n = 15) and *CO1* (n = 4) compared with the others. The overall subpopulation of *ITS2* (Hd = 0.611 ± 0.067) showed lower haplotype diversity than *CO1* (Hd = 0.702 ± 0.031). The summaries of the *CO1* and *ITS2* diversities are shown in Appendix A, respectively. However, *ITS2* (*p* = 0.00165 ± 0.00035) showed higher nucleotide diversity than did *CO1* (*p* = 0.00146 ± 0.00013) for the overall population.

A greater number of haplotypes were found in the *ITS2* (n = 21) and *CO1* (n = 16) sequences of *An. introlatus*. The highest haplotype diversity was noticed in Sg. Sendat, Selangor (Hd = 0.833 ± 0.222), while the highest nucleotide diversity was in Kem Microwave, Johor (*p* = 0.00563 ± 0.00085) for the *CO1* analysis. Among the subpopulations in Hulu Kalong and Sg. Sendat from Selangor, the *ITS2* sequences depicted the highest haplotype diversity (Hd = 1.000 ± 0.126). Similarly, Hulu Kalong had the highest nucleotide diversity (*p* = 0.00513 ± 0.00124).

Compared with *ITS2* (n = 16), *CO1* showed a smaller number of haplotypes, n = 14 for *An. latens*. Based on *CO1*, the Rumah Sewa Panto, Sarawak, subpopulation had the highest haplotype and nucleotide diversities (Hd = 1.000 ± 0.177, *p* = 0.01997 ± 0.00871). In addition, the *ITS2* subpopulation of Rumah Sewa Panto, Sarawak, indicated high nucleotide diversity, *p* = 0.00674 ± 0.00358, but high haplotype diversity was observed in Danum Valley Field Centre, Sabah (Hd = 0.956 ± 0.033).

A total of 16 and 10 haplotypes were observed in the *CO1* and *ITS2* sequences, respectively, for *An. balabacensis*. Based on *CO1*, the highest haplotype diversity was observed in Simpang Utong, Sarawak (Hd = 1.000 ± 0.272), and the highest nucleotide diversity was observed in Lipasu Lama, Sabah (*p* = 0.00200 ± 0.00094). The highest haplotype and nucleotide diversities in *ITS2* were recorded in Limbuak Laut, Sabah (Hd = 0.939 ± 0.058, *p* = 0.00395 ± 0.00080).

### 3.2. Demographic Analyses

The low values of the raggedness index and R_2_ statistic from the mismatch distribution tests and the results of the neutrality test indicated that all the vector species studied were expanding. This expansion trend was further supported by the unimodal shape of the graph observed in at least one gene dataset of the mismatch distribution analysis. These findings collectively suggest that the vector populations are undergoing growth and expansion (Figure 2). Despite a multimodal shape being observed in *An. latens* due to the presence of two distinct lineages, a separate analysis was conducted for each lineage, and a unimodal shape was also observed (unpublished data The mismatch distribution graphs for *CO1* and *ITS2* of *An. introlatus*, *An. latens*, *An. cracens*, and *An. balabacensis* are shown in Appendix A.

### 3.3. Haplotype Network

*An. introlatus* from Johor had the highest number of haplotypes for *CO1*, *ITS2*, and the combined dataset (red colour). A higher number of haplotypes (n = 25) was observed in the combined network, with the majority originating from Johor. *An. latens* from Peninsular Malaysia were clustered distantly from Malaysia Borneo in all three *CO1*, *ITS2*, and combined network analyses. H4 in *An. cracens* held all the populations from Perlis, and the other three haplotypes included the Pahang population for *CO1*. Only H1 (the predominant haplotype) shared populations from Perlis and Pahang for *ITS2*. The haplotypes from the combined network were connected to each other, as they were from the same population (Pahang) of *An. cracens*. The “star-like” shape was observed across all four species. Overall, the combined network for *An. balabacensis* indicated fewer haplotypes (n = 5) compared with *CO1* and *ITS2* (Figure 3).

### 3.4. Genetic Differentiation (F_ST_) and Gene Flow (Nm)

Overall, in the 10 subpopulations, the highest genetic differentiation was observed between Kg. Sg Dara, Perak, and Hutan Lenggor, Johor, for the concatenated sequences of *CO1* and *ITS2* in *An. introlatus* (Table 2). Two out of the three subpopulations exhibited high levels of genetic differentiation, and moderate gene flow was detected between most of the subpopulation pairs in *An. latens* (Table 3). Intermediate genetic differentiation and correspondingly high gene flow were observed between the two subpopulations of *An. cracens* (Table 4). In contrast, low levels of genetic differentiation and gene flow were found between the Kem Kayu Merarap, Sarawak, and Simpang Utong, Sarawak, subpopulations in *An. balabacensis* (Table 5).

## 4. Discussion

This study represents the first attempt to investigate four simian malaria vectors (*An. balabacensis*, *An. cracens*, *An. introlatus*, and *An. latens*) collected from various sites in Malaysia. Based on the combined sequences of *An. introlatus*, the samples from Hulu Kalong and Kongsi Balak had the highest genetic diversities among the other subpopulations, suggesting greater genetic variation and potential population differentiation in these specific locations. The population from Perak had zero diversity for all the sequences analysed. This is likely linked to the lack of variation within the population in Perak [40], sampling errors [24], or the low number of samples (n = 5) within the same area. The presence of the same haplotypes in the different subpopulations, such as in *CO1*-H2, *ITS2*-H1, and the combined sequences-H2, creates the possibility of inter-breeding and migration among the *An. introlatus* subpopulations [24]. Hence, a different number of haplotypes and unique haplotype network structures for each gene can be observed.

The *Anopheles latens* sequences from West and East Malaysia were examined in this study. High haplotype and nucleotide diversities were observed in the subpopulations of East Malaysia (Sabah and Sarawak) for *CO1*, *ITS2*, and the combined sequences. This hypothesis needs further attention because low genetic diversity is the typical characteristic of insects in island populations [41,42,43]. The haplotypes of *An. latens* were separated into two clusters for all three haplotype networks. The haplotype clusters contained sequences from Peninsular Malaysia separated from sequences from Malaysia Borneo. Consequently, it is unknown whether the behaviour and the capability of spreading the simian malaria parasites of these two clusters of *An. latens* mosquitoes are similar or different.

Additionally, this study revealed the presence of the two genetically distinct *An. latens* clusters based on the *CO1* and *ITS2* sequences. Nonetheless, defining a species in *Anopheles* is challenging, even if it is well studied. Thus, multiple genes and concatenated markers were used to improve the accuracy of the assessments of the genetic population structure [44,45]. It is known that *CO1* is commonly used for genetic studies of malaria vectors [46,47,48,49], and *ITS2* is the well-known molecular marker for species identifications [50]. Thus, based on the results obtained from these gene markers, it is plausible to consider the existence of two different types of *An. latens* in Malaysia.

The genetic distances between the two clades of *An. latens* from Peninsular Malaysia and Malaysia Borneo were relatively high, ranging from 2.3% to 5.2% for *CO1* and from 2.3% to 4.7% for *ITS2*. Notably, a cutoff point of 3% is often used as a threshold for species boundaries in insects. Given that the genetic distances observed between the *An. latens* populations from these different geographical regions surpassed this threshold, it strongly suggests that they may represent two distinct species. A detailed morphological examination is also warranted to confirm its species status.

The findings demonstrated that geographic distance might have a major effect on the genetic structure of *An. latens* from the two different geographical regions. The South China Sea significantly divides East and West Malaysia, and this may be a factor leading to the intra-specific genetic discontinuities [51,52]. Therefore, the South China Sea is likely a barrier to gene flow between *An. latens* from Peninsular Malaysia and Borneo. This is further supported by the high levels of genetic differentiation and moderate gene flow in this study.

Overall, the high haplotype diversity observed in *An. cracens* in Pahang indicates its population growth, likely resulting from the accumulation of new mutations over time, with the emergence of new haplotypes within the population. Differences were observed between the *CO1* and *ITS2* markers, which can be attributed to the selection of different molecular markers. Each marker may have distinct mutation or evolutionary rates, varying selection pressures, and different gene constraints [53,54]. Nevertheless, this diversity provided valuable insights into the genetic variation, population structure, and evolutionary processes of the mosquito populations. High gene flow was observed between the Kem Sri Gading and Sg. Ular subpopulation pairs from Pahang, despite being collected from different sites. This can be attributed to the absence of significant geographical barriers between these locations. Furthermore, moderate and low levels of genetic differentiation were observed between the subpopulation pairs. The Sg. Ular subpopulation, located within a durian plantation, and the Kem Sri Gading subpopulation, designated as a reserved forest for camping, tracking, and cycling activities, exhibited distinct ecological characteristics. The association between the geographic distance and the anopheline population genetic structure likely differed by species, most likely due to variations in breeding sites, breeding patterns, and behaviour [55]. The gene flow or genetic differentiation of *An. cracens* in the study were not dependent on the geographical distances. Gene flow among the vectors of malaria due to the lack of geographical distance and barriers was observed in several studies [55,56,57,58,59].

The samples from different study sites could belong to the same, single continuous population because of the low genetic differentiation detected among the subpopulation pairs in both genes for *An. balabacensis*. Nevertheless, low levels of gene flow were also observed from the overall subpopulations in the combined sequences, yet high gene flow was seen in the *COI* and *ITS2* analyses. Although the sampling distributions varied, the ecological habit is relatively similar in Sabah and Sarawak [60]. Thus, the gene flow occurred easily without significant physical barriers.

The “star-like” network observed in all four vectors suggests population expansion [61,62]. Furthermore, the unimodal shape and low values of the raggedness index and R_2_ statistic from the mismatch distribution tests, along with the negative values from the neutrality tests, further support the population expansion of *An. balabacensis*, *An. introlatus*, *An. latens*, and *An. cracens* in Malaysia. Likewise, evidence of demographic expansion has also been reported in other insects, such as dragon flies, black flies, and buffalo flies, in Southeast Asia [63,64,65].

## 5. Conclusions

A deeper understanding of the genetic diversity of local vectors could provide valuable information for epidemiological surveillance and malaria vector control strategies. The presence of the different lineages in the *An. latens* populations could be a topic of interest for future studies, as it would allow for the identification of which genotypes are more likely to be exposed to simian malaria infection in the wild. Furthermore, in this study, the collection sites covered most of the *P. knowlesi* malaria endemic regions in Malaysia, yet future research should conduct extensive sample collection in a wider distribution area to obtain a complete overview of the genetic structure of the vectors. Thus, in light of the elimination of malaria, it is timely for Southeast Asian nations to commit a concerted effort to study the vectors and to develop vector control strategies to prevent future outbreaks.

## Figures and Tables

**Figure 1 genes-14-01369-f001:**
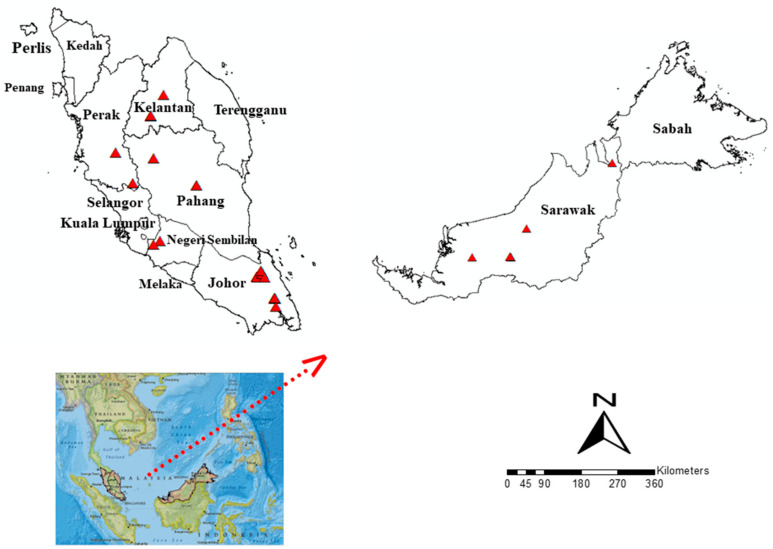
*Anopheles* mosquito collections in Malaysia encompassed various states, spanning both Peninsular Malaysia and Malaysia Borneo.

**Figure 2 genes-14-01369-f002:**
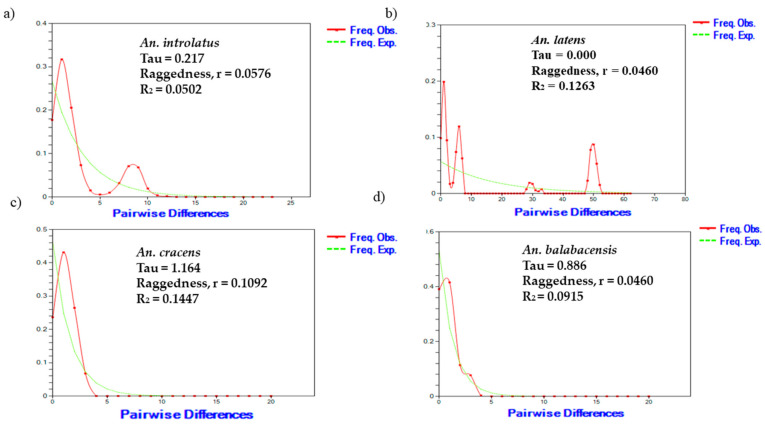
Graphs of the mismatch distribution analysis for (**a**) *An. introlatus*, (**b**) *An. latens*, (**c**) *An. cracens*, and (**d**) *An. balabacensis* based on combined sequences of *CO1Th* and *ITS2*. A multimodal shape was observed in *An. latens* due to the presence of two distinct lineages.

**Figure 3 genes-14-01369-f003:**
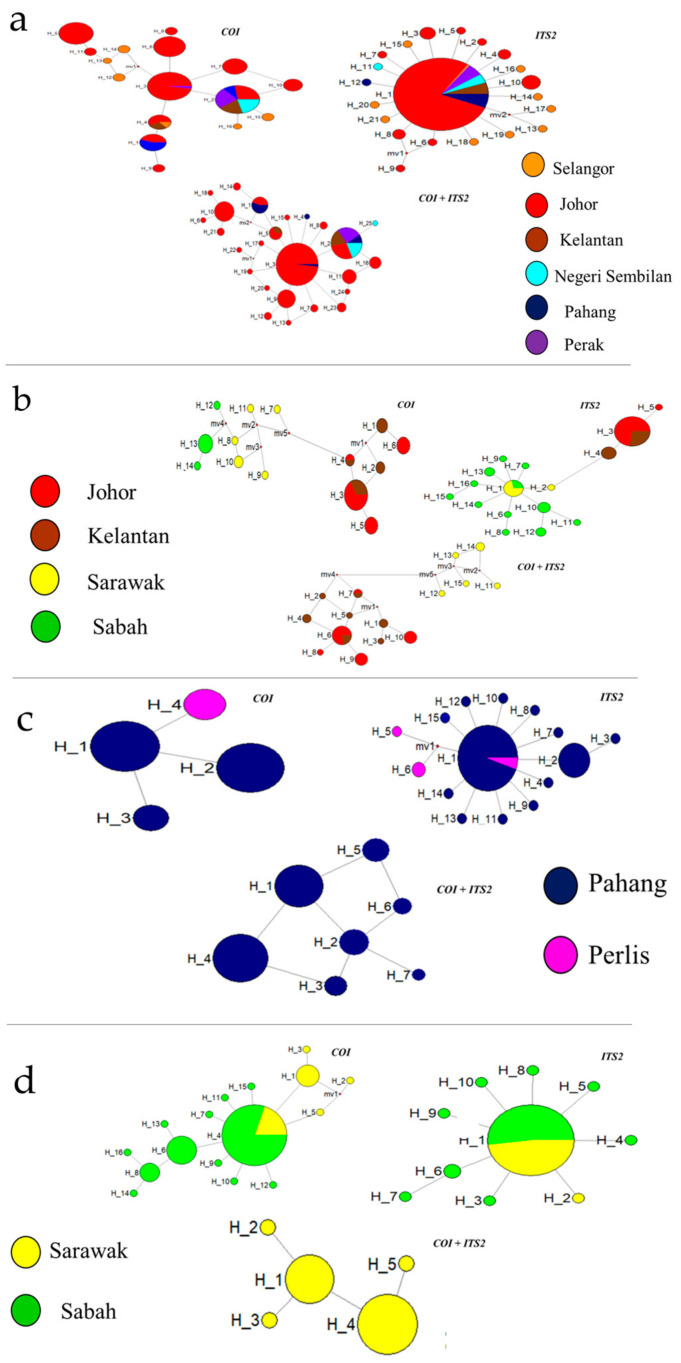
Median-joining networks of (**a**) *An. introlatus*, (**b**) *An. latens*, (**c**) *An. cracens*, and (**d**) *An. balabacensis*.

**Table 1 genes-14-01369-t001:** Summary of concatenated sequences of *CO1* and *ITS2* diversity and neutrality tests in *An. introlatus*, *An. latens*, *An. cracens*, and *An. balabacensis*. Values marked with an asterisk indicate significance: * *p* < 0.05.

Species	Subpopulation	No. of Haplotypes, H/Samples	Nucleotide Diversity, π	Haplotype Diversity, Hd	Neutrality Test	
					Tajima’s D	Fu’s Fs
*An. introlatus*	JOHOR					
	Gunung Panti	3/7	0.00205 ± 0.00206	0.524 ± 0.209	−1.610 *	2.091 *
	Kg. Seri Delima	1/1	-	-	-	-
	Kem Microwave	12/44	0.00266 ± 0.00038	0.810 ± 0.046	−0.645	−1.084
	Hutan Lenggor	9/44	0.00129 ± 0.00036	0.673 ± 0.070	−1.910 *	−2.702
	Kongsi Balak	14/35	0.00273 ± 0.00045	0.899 ± 0.031	−0.429	−3.199
	Kg. OA Punjut	1/1	-	-	-	-
	Kg. OA Berasau	1/1	-	-	-	-
	Total	23	0.00213 ± 0.00024	0.787 ± 0.033	−1.170	−7.773
	KELANTAN					
	Kg. Lalang	2/6	0.00072 ± 0.00032	0.600 ± 0.215	−1.233	−0.189 *
	Kg. Dusun Durian	1/1	-	-	-	-
	Kg. Lebur Jaya	1/1	-	-	-	-
	Total	2	0.00092 ± 0.00024	0.714 ± 0.123	0.458	0.671
	PAHANG					
	Kem Sri Gading	4/10	0.00127 ± 0.00018	0.711 ± 0.117	0.988	0.334
	Total	4	0.00127 ± 0.00018	0.711 ± 0.117	0.988	0.334
	PERAK					
	Kg. Sg. Dara	1/5	0.00000 ± 0.00000	0.000 ± 0.000	0.000	0.000
	Kg. Draco	1/3	0.00000 ± 0.00000	0.000 ± 0.000	-	-
	Total	1	0.00000 ± 0.00000	0.000 ± 0.000	0.000	0.000
	NEGERI SEMBILAN					
	Kebun Durian Tekir	1/2	0.00000 ± 0.00000	0.000 ± 0.000	-	-
	Hutan Lenggeng	2/5	0.00029 ± 0.00017	0.400 ± 0.237	−0.772	0.090
	Total	2	0.00020 ± 0.00014	0.286 ± 0.196	−1.006	−0.095
	Overall Total	25	0.00199 ± 0.00021	0.823 ± 0.022	−1.320	−9.677
*An. latens*	JOHOR					
	Gunung Panti	5/17	0.00267 ± 0.00052	0.824 ± 0.064	0.676	−0.223
	Total	5	0.00267 ± 0.00052	0.824 ± 0.064	0.676	−0.223
	KELANTAN					
	Kg. Lalang	7/10	0.00255 ± 0.00041	0.933 ± 0.062	1.758	−2.029
	Total	7	0.00255 ± 0.00041	0.933 ± 0.062	1.758	−2.029
	SARAWAK					
	Taman Ixora	1/1	-	-	-	-
	Kg. Sawang	1/1	-	-	-	-
	Rumah Sewa Panto	4/4	0.01410 ± 0.00642	1.000 ± 0.177	−0.566	0.903
	Total	5	0.01037 ± 0.00510	0.933 ± 0.122	−1.177	1.138
	Overall Total	15	0.01419 ± 0.00344	0.902 ± 0.036	0.195	2.674
*An. cracens*	PAHANG					
	Kem Sri Gading	7/37	0.00093 ± 0.00009	0.791 ± 0.044	0.800	−1.573
	Sg. Ular	2/8	0.00039 ± 0.00009	0.536 ± 0.123	1.167	0.866
	Overall Total	7	0.00085 ± 0.00008	0.764 ± 0.040	0.632	−1.600
*An. balabacensis*	SARAWAK					
	Kem Kayu Merarap	4/21	0.00065 ± 0.00018	0.557 ± 0.092	−0.848	−0.521
	Simpang Utong	3/3	0.00168 ± 0.00059	1.000 ± 0.27	-	-
	Kebun Ldg Sawit Jelapang	1/2	0.00000 ± 0.00000	0.000 ± 0.000	-	-
	Overall Total	5	0.00074 ± 0.00017	0.609 ± 0.068	−0.917	−1.106

**Table 2 genes-14-01369-t002:** Genetic differentiation (*F_ST_*) and gene flow (Nm) between subpopulations of *An. introlatus* based on concatenated sequences of *CO1* and *ITS2*. Values above the diagonal are for Nm, while values below the diagonal are for *F_ST_*. Values marked with an asterisk indicate that the genetic distances between two subpopulations are significant: * *p* < 0.05, ** *p* < 0.01, *** *p* < 0.001.

Subpopulation		1	2	3	4	5	6	7	8	9	10
Gunung Panti	1	-	−6.680	−9.350	−10.350	0.710	1.950	0.360	0.290	0.370	0.410
Kem Microwave	2	−0.039	-	3.670	12.750	0.710	2.280	0.360	1.620	1.490	0.400
Hutan Lenggor	3	−0.027	0.064 *	-	5.470	0.610	1.130	0.240	1.380	1.350	0.300
Kongsi Balak	4	−0.025	0.019 *	0.044 *	-	1.210	1.550	0.610	1.780	1.590	0.670
Kg. Lalang	5	0.260 *	0.260 *	0.289 **	0.171	-	0.680	2.750	3.750	3.600	2.450
Kem Sri Gading	6	0.114 *	0.099 *	0.181 ***	0.139 *	0.269	-	0.240	0.980	1.100	0.300
Kg. Sg Dara	7	0.412 *	0.412 ***	0.508 ***	0.291	0.049	0.506	-	≠	≠	2.000
Kg. Draco	8	0.464 *	0.134 **	0.153 **	0.123	0.049	0.203	≠	-	≠	2.670
Kebun Durian Tekir	9	0.403 *	0.144 *	0.157 ***	0.136	0.064	0.186	≠	≠	-	3.500
Hutan Lenggeng	10	0.380 *	0.387 ***	0.458 *	0.271	0.033	0.456	0.111	0.008	0.031	-

≠ This sign indicates polymorphic sites were not observed in the selected subpopulation pairs.

**Table 3 genes-14-01369-t003:** Genetic differentiation (*F_ST_*) and gene flow (Nm) among the subpopulations of *An. latens* based on the concatenated sequences of *CO1* and *ITS2*. Values above the diagonal are for Nm, while values below the diagonal are for *F_ST_*. Values marked with an asterisk indicate that the genetic distances between the two subpopulations are significant: * *p* < 0.05.

Subpopulation		1	2	3
Gunung Panti	1	-	4.980	0.070
Kg. Lalang	2	0.048	-	0.070
Rumah Sewa Panto	3	0.785 *	0.780	-

**Table 4 genes-14-01369-t004:** Genetic differentiation (*F_ST_*) and gene flow (Nm) between the subpopulations of *An. cracens* based on the concatenated sequences of *CO1* and *ITS2*. Values above the diagonal are for Nm, while values below the diagonal are for *F_ST_*.

Subpopulation		1	2
Sg. Ular	1	-	3.150
Kem Sri Gading	2	0.074	-

**Table 5 genes-14-01369-t005:** Genetic differentiation (*F_ST_*) and gene flow (Nm) between the subpopulations of *An. balabacensis* based on the concatenated sequences of *CO1* and *ITS2*. Values above the diagonal are for Nm, while values below the diagonal are for *F_ST_*.

Subpopulation		1	2	3
Kem Kayu Merarap	1	-	−2.280	0.260
Simpang Utong	2	−0.123	-	0.750
Kebun Ldg Sawit Jelapang	3	0.488	0.250	-

## Data Availability

All data are available within the manuscript.

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
