# Peer review of "Cryptic Diversity and Demographic Expansion of Plasmodium knowlesi Malaria Vectors in Malaysia"

_genes, 2023, doi:10.3390/genes14071369_

Round 1

Reviewer 1 Report

The authors present results on population genetic structure and genetic diversity of four important knowlesi malaria vectors in Malaysia.

I believe that in this current form, the study is not suitable for publication. Some major issues are presented bellow.

Introdution: The authors fail to provide pivotal information on the importance of the simian malaria parasites as the emerging cause of malaria in humans. Thers a lot of crucial information that is missing in the manuscript that can be easily found elsewhere (i.e. Sam et al., 2022)

M and M: the map of study location is poorly described. What are those two separated regions? please provide a map that one can see all Malysia and then the region of study

Results and Discussion:  The figure showing the haplotype tree should be presented separated, based on each species. It is virtually impossible to read.

There is no p value at all, describing the statistics for Fst and Nm. Without those values it is pointless to discuss whether it was found a high or low value for the genetic tests.

The numbers of found hapotypes presented in the text do not match the numbers discribed in the table 1. For instance, in the text the authors mention that for An. balabacen A total of 16 and 10 haplotypes were observed in COI and ITS2 sequences, but in the table the overall haplotypes presented are only 5!. Moreover the authors do not discuss the results for the neutrality tests and again do not present pvalue for the comparisons.

Moderate english review is suggested

Author Response

All the points raised by the reviewer 1 has been answered in the attached file

Reviewer 2 Report

The manuscript is not appropriate for acceptance in current form. Major edits required. Studies like these with appropriate explanation are necessary and will contribute to the field of malaria surveillance.

Major points/questions:

1.     The sequencing approach/technology used is not mentioned in the paper and there is no way to judge the quality of the data, as no estimates of coverage or depth are provided. The quality of data will affect the results greatly.

2.     This manuscript lacks information on data sampling. How many samples were collected/sequenced from each location? This is important to know as this also will have a great impact on diversity calculations.

3.     When the sequences of both CO1 and ITS2 are recovered for all the samples, what is the need of doing separate and combined analysis? There is no justification of the approach provided and additional repetitive analysis is not adding any additional information to support the conclusion.

4.     How are the subpopulations described? No subpopulation analysis was performed. If these subpopulations are described just based on the sampling location, why is it not mentioned in the introduction and methods section?

5.     The conclusions are not entirely supported by the results. Some of the necessary estimates like linkage disequilibrium and genetic distance are mentioned without any explanations or results. The conclusions are exaggerated without providing any alternative possibilities. The results do not support some of the mentioned genome evolution context based conclusions.

Minor points/suggestions:

1.     Title:

a.     I suggest adding “Plasmodium knowlesi” to the title.

2.     In section 2.1,

a.     In line 87, add Plasmodium to the name when referring to P. knowlesi.

3.     In Figure 1,

a.     Mentioning the number of samples from each province and the location of the subpopulations will help in understanding and validating the results.

4.     In section 2.3,

a.     What sequencing technology was used?

5.     In section 2.4,

a.     Provide reference for BioEdit tool.

b.     In line 136, in R2, 2 should be subscript.

6.     In section 3.1,

a.     In line 149-150, what is the significance of this difference in nucleotide diversity?

7.     In section 3.2,

a.     In line 172-174, explain this conclusion in detail. How Rag, R2 and Tajima’s D explain the expansion of vector populations?

8.     In Table 1,

a.     In line 179, keep notation of CO1 consistent throughout the manuscript.

b.     How are the sub-populations described. It is important to mark the subpopulations on the map as well with number of samples.

9.     In Figure 2,

a.     Why do we see multimodal distribution (except An. cracens)? Explain the observation and result.

b.     The legends and axis should be bigger.

10.  In Figure 3,

a.     High quality images should be provided with proper aspect ratio.

11.  In section 3.3,

a.     In line 205, How are the subpopulations defined?

b.     In line 209, wrong table referenced.

c.     In line 210-214, rewording required.

12.  In Table 2,

a.     Which FST estimate has been used? The range of FST is 0 to 1.

b.     Mention range of gene flow.

13.  In section 4,

a.     In line 255-256, What do you mean by “suggesting a good maintenance of genetic material within the population”. Explain.

b.     In line 258, lack of variation between what populations?

c.     In line 272, are the capabilities for other Anopheles species known?

d.     In line 278, How is the genetic distance calculated?

e.     In line 281, How do you conclude that these could be cryptic species?

f.      In line 291, How were the subpopulations defined?

g.     In line 296, No linkage has been assessed. Show the results if it has been.

h.     In line 297-298, How haplotype diversity and nucleotide diversity suggesting population growth?

i.      In line 316-317, How is the star-like structure suggesting population expansion?

j.      In line 317-320, mismatch distribution is not unimodal and how does it support the population of different species. Rewording and detailed explanation required.

14.  In section 5,

a.     In line 6, how close is An. maculatus to simian An. species?

In some sections the sentences are unnecessarily long and it is difficult to follow the manuscript. At many places the statements are incomplete and are not scientifically sound.

Author Response

All the comments have been answered. It is in the attached file. 

Round 2

Reviewer 1 Report

The authors have improved the original version of the MS and answered all the issues raised during the review process. Therefore, i feel confortable to recommend the MS for publication.

Reviewer 2 Report

The manuscript is appropriate for acceptance with minor edits.

Minor points/suggestions:

1.     In Figure 3

a.     Provide figures with correct aspect ratio. Each haplotype should be represented with a “circle” (not oval, as represented in some cases)

b.     A legend for circle size should be made available.